# What Really Matters in Matrix-Whitening Optimizers?

## Abstract

A range of recent optimizers have emerged that approximate the same *matrix-whitening* transformation in various ways. In this work, we systematically deconstruct such optimizers, aiming to disentangle the key components that explain performance. Under tuned hyperparameters across the board, all flavors of matrix-whitening methods reliably outperform their elementwise counterparts, such as Adam. Matrix-whitening is often related to spectral descent – however, metrics reveal that performance gains are *not explained solely by accurate spectral normalization* – particularly, SOAP displays the largest per-step gain, even though Muon more accurately descends along the steepest spectral descent direction. Instead, we argue that matrix-whitening serves *two* purposes, and the *variance adaptation* component of matrix-whitening is the overlooked ingredient explaining this performance gap. Experiments show that variance-adapted versions of optimizers consistently outperform their sign-descent counterparts, including an adaptive version of Muon. We further ablate variance adaptation strategies, finding that while "lookahead" style approximations are not as effective, low-rank variance estimators can effectively reduce memory costs without a performance loss.

## 1 Introduction

In recent years, increasing growth in the scale of neural networks has resulted in a strong need to understand how neural networks can be trained efficiently. The workhorse of modern deep learning, gradient descent, has proven extensively scalable yet remains an inherently iterative process. By gaining a deeper understanding of such processes through both theoretical and empirical reconciliation, the field may continue the steady march in improving neural network training.

A range of recent optimizers have emerged that share a similar *matrix-whitening* transformation (Yang & Laaksonen, 2008; Carlson et al., 2015b; Gupta et al., 2018). While differing in their exact approximations and implementation, such optimizers can generally be derived from the same core principles (Bernstein & Newhouse, 2024). However, the concrete algorithms proposed have often contained auxiliary implementation details, potentially obscuring the root cause of the performance gain. Clarity is at times obscured further by uneven hyperparameter tuning (Schmidt et al., 2021; Zhao et al., 2024; Wen et al., 2025).

In this work, we systematically deconstruct such optimizers, aiming to disentangle the key components that explain performance. We establish a controlled experimental setup, with an explicit emphasis on breaking down methods into their constituent parts. We conduct a thorough sweep over four key hyperparameters, noting that optimal learning rate and weight decay parameters vary greatly across optimizer flavors. When all methods are tuned, we confirm that matrix-whitening optimizers reliably outperform elementwise transformations like Adam by a nontrivial margin.

However, the story *comparing* matrix-whitening optimizers is less clear. Empirically in our setting, SOAP (Vyas et al., 2024) displayed the largest per-step gain in performance, outperforming Muon (Jordan et al.). In an effort to understand the cause of these gains, we consider a hypothesis that the strength of matrix-whitening comes from its interpretation as steepest spectral descent (Bernstein & Newhouse, 2024), and that Shampoo-style explicit matrix inversion provides a more accurate spectral normalization than approximate Newton-Schulz iteration. However, metrics show that Muon-style methods result in a *tighter* spread of singular values than SOAP, leading to a conclusion that *performance gains are not explained solely by accurate spectral normalization*.

In contrast, we argue that matrix-whitening serves *two* purposes—both spectral normalization and *variance adaptation*, and this variance adaptation aspect of whitening is a crucial, and often overlooked, ingredient in achieving strong performance. We identify three optimizer pairs that utilize the same spectral transformation, but opt to use signed descent vs. variance-adapted descent – Signum vs. Adam, SPlus vs. SOAP, and Muon vs. AdaMuon. In all cases, the variance-adapted versions result in superior performance difference *almost equal to the gap between Adam and Muon*.

Having understood the above relationship, we then seek to understand how gains from variance-adaptation can be achieved with less computational and hyperparameter requirements. We begin by considering the family of "lookahead" optimizers that can be seen as approximating a continuous function over expectations over the sign, but conclude that this is ineffective in closing the gap. Instead, we show that low-rank approximations of elementwise variance estimates can be used with negligible impact, at at times even superior performance.

Our main contributions in this work are in establishing a controlled experimental framework for comparing optimizer flavors, and in the use of this framework to identify variance-adaptation as an critical ingredient. We further detail this claim through a thorough ablation of variance-adaptation across three matrix-whitening optimizer families. We *do not claim* that the spectral-descent view of matrix-whitening is incorrect, rather, we show that spectral normalization is consistently effective, but argue it is not the full picture, and pure orthogonalization methods – such as Muon, Dion (Ahn et al., 2025) and Polargrad (Lau et al., 2025), among others – can be further improved. We hope our findings encourage the study of optimizer flavors in terms of interchangeable components rather than entirely separate methods.

## 2 RELATED WORK

**Optimization for neural networks.** The search for strong neural network optimization strategies has a long history alongside the adoption of deep learning (LeCun et al., 2002; Martens et al., 2010; Sutskever et al., 2013). Two crucial techniques are momentum along with adaptive elementwise preconditioning (Hinton, 2012; Duchi et al., 2011), which are combined in the Adam optimizer (Kingma & Ba, 2014). Adam can seen as an elementwise approximation to the *whitening metric* (as discussed in Equation (2)), a metric which has also been related to second-order descent over the Hessian (in particular, the Gauss-Newton approximation) (Martens et al., 2010; Korbit et al., 2024; Bottou et al., 2018; Schraudolph, 2002; Li, 2017; Pooladzandi & Li, 2024; LeCun et al., 2002; Liu et al., 2023), to natural gradient descent over a form of the Fisher information matrix (Amari, 1998; Sohl-Dickstein, 2012; Kunstner et al., 2019), to a signal-to-noise trust region (Balles & Hennig, 2018; Orvieto & Gower, 2025), and to descent under the spectral norm (Bernstein & Newhouse, 2024). Our work takes a step towards bridging these perspectives, showing how performance may in fact come from several of these arguments, as we will show concretely.

**Matrix-based optimizers.** More recently, optimizers have been proposed that explicitly account for the matrix-based structure of neural networks. K-FAC (Martens & Grosse, 2015) introduced a dimension-wise Kronecker factorization scheme, which was further refined in Shampoo (Gupta et al., 2018) and its variants. PSGD (Li, 2017; 2018) also utilizes this scheme in its Kron variety. The Muon family (Jordan et al.) again utilizes the matrix-structure of a network to define a computationally efficient orthogonalization procedure, with alternate orthogonalization techniques being an open problem (Ahn et al., 2025; Lau et al., 2025). Our work views such methods in a unified light, allowing experiments at the resolution of individual *components* of matrix-based optimization.

**Benchmarks for neural network training.** There have been a number of previous works which evaluate a suite of optimizers for comparison purposes (Schmidt et al., 2021; Dahl et al., 2023; Kasimbeg et al., 2025; Kaddour et al., 2023), some of which are concurrent (Wen et al., 2025; Semenov et al., 2025). Closest to our work in flavor are Zhao et al. (2024) and Wen et al. (2025), which similarly place an emphasis on disentangling performance via careful sweeping of hyperparameters, with the latter considering matrix-based optimizers. The difference is that in this work, we explicitly control for all auxiliary decisions (e.g., the optimization strategy for non-matrix parameters) and deconstruct each optimizer into only its minimal transformation – this allows us to conduct fine-grained ablations, eventually concluding that the spectral-normalization aspect of matrix-whitening may not be the full story.

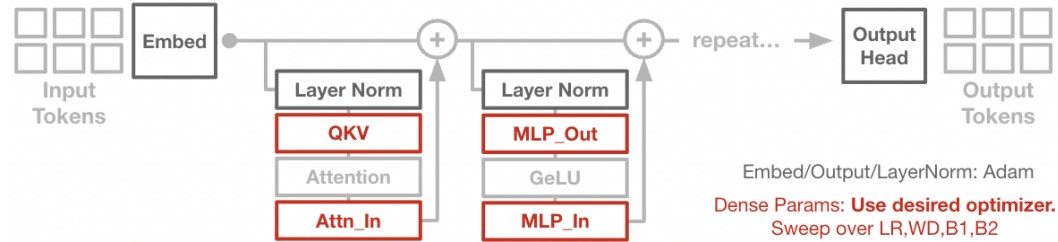

Figure 1: **Our experimental setup aims to isolate the core effects of various matrix-whitening optimizers on Transformer training.** For each method, we sweep over learning rate, weight decay, $\beta_1$, and $\beta_2$. All runs use the same initial parameters and data ordering. Nonstandard parameters (embed, output, and layernorm) are optimized using Adam with fixed tuned hyperparameters.

## 3 PRELIMINARIES

**Gradient descent on non-Euclidean metrics.** Gradient descent can be seen as solving for a trade-off between linear improvement and a distance penalty over parameters. While standard gradient descent assumes a Euclidean distance over parameters, we can generally represent second-order distances using a symmetric positive-definite metric $M$, with an analytic solution of:

$$u = \operatorname{argmin}_{\Delta\theta} \underbrace{-g^T\Delta\theta}_{\text{Improvement}} + \underbrace{(1/2)\Delta\theta^T M\Delta\theta}_{\text{Distance Penalty}} = M^{-1}g, \tag{1}$$

where the matrix-inverse $M^{-1}$ is sometimes referred to as a *preconditioner*.

**Whitening metric.** While there are many possible distance metrics to descend on, many recent optimizers have converged on a specific metric in particular, which we refer to as the *whitening* metric following (Yang & Laaksonen, 2008). Mechanically, the whitening metric can be written as the square-root uncentered covariance of incoming gradients:

$$M_{\text{Whitening}} = \mathbb{E}_{x,y}\left[\nabla_\theta L(\theta, x, y)\nabla_\theta L(\theta, x, y)^T\right]^{1/2} = \mathbb{E}_{x,y}\left[gg^T\right]^{1/2}. \tag{2}$$

Prior works have examined the relation of the whitening metric to the Hessian and to the Fisher information matrix, for which we defer to previous discussion (Kunstner et al., 2019). Adam (Kingma & Ba, 2014) can be understood as utilizing an *elementwise* approximation to the whitening metric, resulting in an efficient update where $m = diag(M)$:

$$m = E_{x,y}\left[g^2\right] \qquad u = g/m. \tag{3}$$

**Matrix-based whitening.** Two powerful connections appear when we accept that in neural networks, parameters are structured *matrices* rather than an arbitrary set. First, we can represent the per-layer whitening metric in terms of its Kronecker factors. For dense layer parameters with the natural matrix form $g \in R^{mn} \leftrightarrow G \in R^{m,n}$, this defines the convenient approximation:

$$gg^T \quad \leftarrow \text{approx.} \rightarrow \quad (GG^T)^{1/2} \otimes (G^TG)^{1/2}. \tag{4}$$

The key benefit of Kronecker approximation is that we can precondition via the inverted Kronecker factors directly, without ever actually forming the full product. This results in the following matrix-form whitening update utilized by the Shampoo (Gupta et al., 2018) family:

$$E_{x,y}[gg^T]^{-1/2}g \quad \leftarrow \text{approx.} \rightarrow \quad E_{x,y}[GG^T]^{-1/4} G \, E_{x,y}[G^TG]^{-1/4} \tag{5}$$

Second, if we ignore the expectation, the term above is equivalent to the *orthogonalization* of $G$ (Carlson et al., 2015b;a; Tuddenham et al., 2022; Bernstein & Newhouse, 2024; Lau et al., 2025). This relation can be derived by rewriting $G$ as its singular-value decomposition, $G = U\Sigma V^T$:

$$(GG^T)^{-1/4} G (G^TG)^{-1/4} = (U\Sigma^2 U^T)^{-1/4} U\Sigma V^T (V\Sigma^2 V^T)^{-1/4} = UV^T, \tag{6}$$

and is the solution to steepest descent under the *spectral norm* of the matrix.

A range of optimizer families – such as PSGD, Shampoo, and Muon – can be seen as approximating the above behaviors, and we refer to these as **matrix-whitening** methods. While similar in motivation, these families differ in their core algorithmic decisions, and we will take a step towards disentangling these choices in the following section.

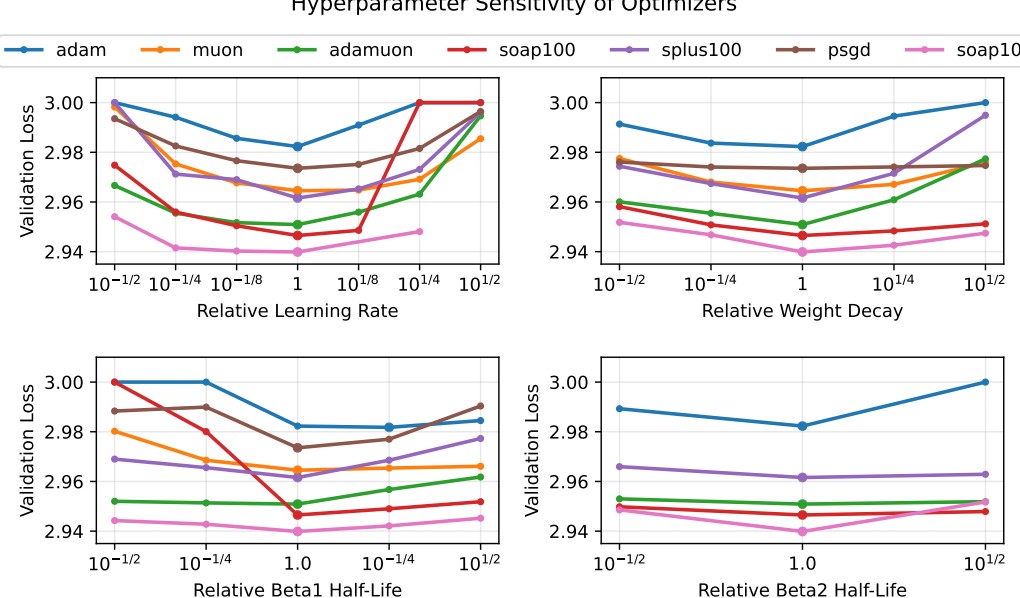

Figure 2: All methods are tuned to within a local optimum of four key hyperparameters. **Matrix-whitening optimizers generally maintain their relative performance gains across local adjustments to hyperparameters.** Plots are centered around each method's optimal hyperparameters.

## 4 EXPERIMENTAL SETUP

We now conduct an empirical study of optimizers that approximate the matrix-whitening update. In our experimental setting (Figure 1), we train a standard GPT-2 architecture Transformer (Brown et al., 2020; Vaswani et al., 2017) on a next-token prediction language modelling objective with the OpenWebText dataset (Gokaslan et al., 2019). The model follows the "Base" size architecture and has 162M total parameters. We train for 10,000 gradient steps on a batch of 1024 sequences of length 256, which is roughly a 1x Chinchilla ratio (Hoffmann et al., 2022). We use a fixed warmup of 200 steps and a cosine learning rate schedule afterwards.

The primary aim of this comparison is to remove confounding factors and examine only the core differences between each optimizer. Thus, we ensure that each trial uses the same data ordering, random seed, and initial parameters. For nonstandard parameters (i.e. layer norm scales and input/output heads), we update using a separate Adam optimizer with fixed tuned hyperparameters. Whenever possible, we disregard auxiliary design choices in each algorithm (e.g. learning rate grafting, Nesterov momentum, or iterate averaging) and focus on the core whitening behavior.

Importantly, we sweep over four key hyperparameters – learning rate, weight decay, momentum coefficient $\beta_1$, and variance coefficient $\beta_2$ (when applicable) – and do so independently for each method. We sweep learning rate within a resolution of $10^{1/8} \approx 1.32$, weight decay within a resolution of $10^{1/4} \approx 1.78$, $\beta_1$ within a half-life resolution of $10^{1/4} \approx 1.78$, and $\beta_2$ within a half-life resolution of $10^{1/2} \approx 3.15$. All methods are tuned to within a local optimum of these hyperparameters as displayed in Figure 2. As discussed in Table 1, we believe this resolution to be sufficient to differentiate performance.

We benchmark the performance of the following optimizers, choosing method-specific settings that lead to the strongest performance when computationally reasonable:

- **Adam** (Kingma & Ba, 2014), a baseline optimizer that is the current standard for training deep neural networks. Updates are normalized by an elementwise second moment buffer.

- **Signum** (Bernstein et al., 2018), which updates via the elementwise sign rather than normalizing by second-moment.

| Val. Loss | LR | WD | $\beta_1$ Half-Life | $\beta_2$ Half-Life | Comparable to... |
|---|---|---|---|---|---|
| $\pm 0.005$ | 1.33 | 1.78 | 1.78 | $\geq 3.15$ | n/a |
| $\pm 0.01$ | 1.78 | 3.15 | 3.15 | $\geq 3.15$ | SOAP-100 vs. SOAP-10 |
| $\pm$ **0.02** | 1.78 | $\geq 3.15$ | $\geq 3.15$ | $\geq 3.15$ | Adam vs. Muon |

Table 1: Required hyperparameter tuning resolution to achieve a desired resolution in validation loss. **We tuned hyperparameters to a resolution of** $\pm 0.005$ **validation loss, enough to distinguish between optimizer flavors which can result in a difference of** $\pm 0.02$.

| Method | LR | WD | $\beta_1$ | $\beta_2$ | Walltime | Adam Steps | Val Loss |
|---|---|---|---|---|---|---|---|
| Adam | 0.001 | 1.0 | 0.95 | 0.99 | 1.0 | 1.0 | $2.982_{\pm.008}$ |
| Signum | 0.000177 | 3.162 | 0.9 | - | 1.0 | $> 1.0$ | $3.006_{\pm.008}$ |
| PSGD | 0.000264 | 0.001 | 0.968 | - | 4.8 | 0.95 - 1.0 | $2.973_{\pm.006}$ |
| Shampoo-100 | (Fails to converge) | | | | | | |
| Shampoo-10 | 0.00132 | 1.0 | 0.9 | 0.99 | 3.2 | 0.80 - 0.83 | $2.963_{\pm.004}$ |
| SPlus-100 | 0.1 | 0.01 | 0.99 | 0.968 | 1.3 | 0.80 - 0.83 | $2.962_{\pm.007}$ |
| SPlus-10 | 0.1 | 0.01 | 0.99 | 0.99 | 3.2 | 0.77 - 0.80 | $2.954_{\pm.007}$ |
| **SOAP-100** | 0.00175 | 0.316 | 0.9 | 0.99 | 1.2 | 0.71 - 0.74 | $\mathbf{2.946}_{\pm.003}$ |
| **SOAP-10** | 0.00311 | 0.316 | 0.968 | 0.99 | 3.1 | 0.66 - 0.68 | $\mathbf{2.939}_{\pm.003}$ |
| Muon | 0.00770 | 0.1 | 0.9 | - | 1.07 | 0.80 - 0.83 | $2.964_{\pm.005}$ |
| **AdaMuon** | 0.000312 | 3.162 | 0.968 | 0.99 | 1.07 | 0.74 - 0.77 | $\mathbf{2.950}_{\pm.003}$ |

Table 2: Under optimal hyperparameters, **matrix-whitening methods outperform Adam**. The highest per-step performance is achieved by SOAP, followed by AdaMuon which strikes a strong balance between wallclock time and final validation loss. "Adam Steps" compares against how long Adam takes to reach an equivalent validation loss, see Appendix Section A.3 for details.

- **Shampoo** (Gupta et al., 2018; Shi et al., 2023), a matrix optimizer which explicitly tracks Kronecker factors as in Equation (5). Every N gradient steps, the left and right preconditioners are calculated by raising each factor to the $-(1/4)$ matrix power, and this result is cached until the next recalculation. We consider $N \in \{10, 100\}$.

- **SOAP** (Vyas et al., 2024), a variant of Shampoo where updates are rotated onto the *eigenbasis* of the left/right factors. In this rotated space, the updates are normalized via an elementwise uncentered variance (i.e. an inner Adam update), then rotated back.

- **SPlus** (Frans et al., 2025), which similarly to SOAP rotates updates onto the eigenbasis, but takes the elementwise sign rather than normalizing by an explicit second moment buffer.

- **Muon** (Jordan et al.), an optimizer which implicitly orthogonalizes updates via Newton-Shulz iteration, and can be seen as descending under the spectral norm (Equation (6)).

- **AdaMuon** (Si et al., 2025), a variant on Muon where a variance buffer is estimated over *post*-orthogonalized updates, and is used for elementwise normalization. We use a simplified form of the original algorithm that does not use the pre-NS sign transformation.

- **PSGD (Fisher-Kron)** (Li, 2017; 2018), which keeps track of a left/right preconditioner that is learned via iterative gradient descent. We update the preconditioner at every step.

For all optimizers, preconditioning is performed on a momentum buffer, as is standard practice.

As shown in Table 2, the considered set of optimizers outperform Adam across the board, reaching an equivalent validation loss within between **66% to 83%** of the gradient steps for the Shampoo and Muon families. We report a margin of error as the difference within our smallest hyperparameter search resolution, and note that the gap between optimizer flavors is an order-of-magnitude higher. Notably, the gains in performance from utilizing a more performant optimizer are consistent even when considering sub-optimal hyperparameters, e.g. Muon with a 2x greater-than-optimal learning rate remains stronger than Adam with the equivalent adjustment, as shown in Figure 2.

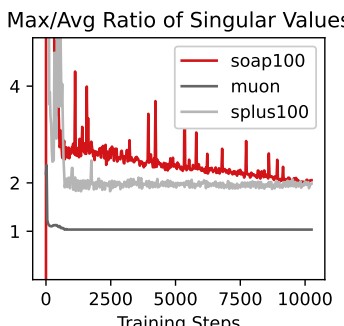
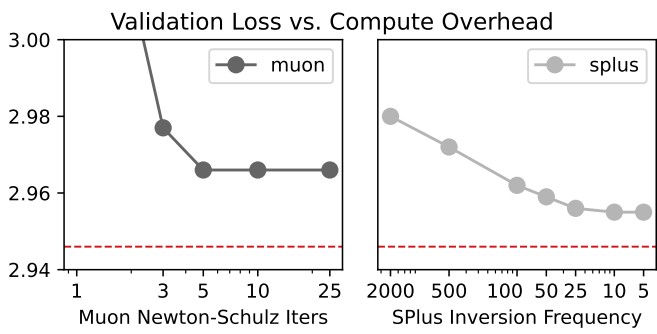

Figure 3: **Left: Muon descends under the spectral norm more accurately than SOAP or SPlus.** This is achieved when all singular values in the update are $\pm 1$, and the ratio between the maximum and average is close to 1. In contrast, the Shampoo-style methods perform this only loosely, with a ratio between 2 to 3. Adam results in a ratio of $\approx 12$ (not plotted). **Right: Even with increased computation, Muon or SPlus do not reach the empirical performance of SOAP.** For Muon, we increase the number of Newton-Schulz iterations at each step. For SPlus, we increase the frequency of updating the eigenbasis. The red dotted line represents the performance of SOAP-100.

## 5 PERFORMANCE GAINS ARE NOT EXPLAINED SOLELY BY ACCURATE SPECTRAL NORMALIZATION

In our experimental setting, SOAP displays the largest per-step gain in performance, and both SOAP-100 and SOAP-10 outperform other optimizer flavors. We have reasonably outruled the hypothesis of unequal hyperparameter tuning. What other reasons may explain the difference?

One notable comparison is between SOAP and Muon, as the two optimizers utilize different computational strategies to perform the matrix-whitening operation. SOAP keeps a historical average of the left and right second moments, then uses an explicit solver to locate the eigenbasis (we use `eigh` in our implementation). Incoming momentum buffers are then rotated onto this basis, normalized elementwise, then rotated back. In contrast, Muon utilizes a Newton-Schulz iteration to implicitly orthogonalize the momentum buffer, aiming to set all singular values to $\pm 1$.

A reasonable hypothesis is that the approximate nature of the Newton-Schulz iteration is not as effective as the explicit eigendecomposition used in Shampoo-style methods. To investigate this claim, we log both the maximum singular value (i.e. spectral norm) and the average singular value of updates, visualized in Figure 3 (left). As expected, the gap between the maximum and average singular values is largest in Adam, around $\approx 12$. However, in comparison to SOAP which ranges from 2 to 3, *Muon achieves a tighter spread in its singular values, with a ratio very close to 1*. In other words, **even though Muon achieves a more accurate solution to the steepest descent direction under the spectral norm (Equation (6)), SOAP results in a stronger final performance**.

Additionally, we show that the eigenbasis pairs of SOAP can be even further approximated, and performance still remains stronger than Muon. First, SOAP utilizes a *cached* eigenbasis for computational reasons, and performance remains strong even when this eigenbasis is cached for 100 gradient steps. Second, it has been shown that SOAP can be performed with only one side preconditioned with relatively little degradation (Vyas et al., 2024). We confirm these claims, and additionally show that *the output basis can be completely ignored* – i.e. the matrices are only rotated along the input axis – and performance is negligibly affected (Appendix Section A.1).

As a final point of evidence, we find in Figure 3 (right) that Muon and SPlus cannot reach the performance of SOAP even with additional computational budget for the optimizer. Specifically, we increase the amount of Newton-Schulz iterations in Muon, and the frequency of matrix-inversions in SPlus, and find that gains from a more accurate preconditioner plateau.

Together, these observations lead us to believe that faithfully descending along the spectral norm may not be the optimal behavior for a matrix-whitening optimizer. Instead, are there other aspects of matrix-whitening that may be equally important?

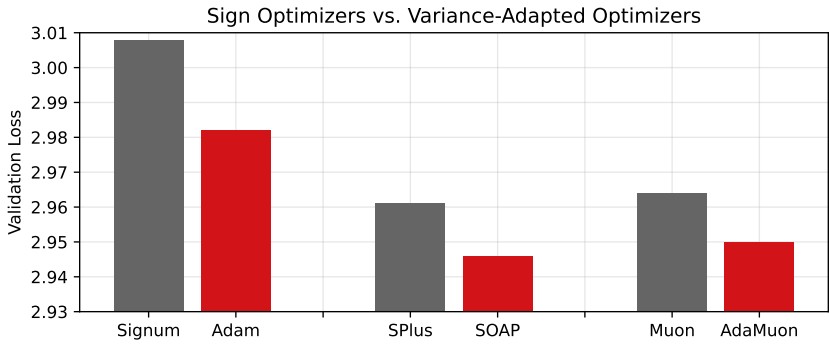

Figure 4: **Variance-adapted variants of optimizers outperform their strictly signed-descent counterparts.** As elaborated more in Table 3, these improvements remain when the variance buffer is factorized into a rank-1 approximation, as well as to a less degree when $\beta_1 = \beta_2$. Variance adaptation can be interpreted as imposing a signal-to-noise dependent adaptive trust region, composable with the rotational or spectral-normalizing aspects of matrix-whitening.

## 6 VARIANCE ADAPTATION IS A CRUCIAL MATRIX-WHITENING INGREDIENT

When examining the design of optimizer flavors, a recurring choice occurs in approximating Equation (2) – regardless of the prior or post transformations, a raw update can be normalized by either 1) its instantaneous sign, or 2) by its square-root historical (uncentered) variance, which we refer to as **variance adaptation** following (Balles & Hennig, 2018). This distinction can be made explicitly clear by considering three pairs of optimizers:

$$\textbf{Signum: } \text{sign}(\bar{g}) \quad \rightarrow \quad \textbf{Adam: } \bar{g} \oslash \mathbb{E}[g^2]^{-1/2} \tag{7}$$

$$\textbf{SPlus: } \text{unrot}(\text{sign}(\text{rot}(\bar{g}))) \quad \rightarrow \quad \textbf{SOAP: } \text{unrot}(\text{rot}(\bar{g}) \oslash \mathbb{E}[\text{rot}(g)^2]^{-1/2}) \tag{8}$$

$$\textbf{Muon: } \text{NS}(\bar{g}) \quad \rightarrow \quad \textbf{AdaMuon: } \text{NS}(\bar{g}) \oslash \mathbb{E}[\text{NS}(g)^2]^{-1/2} \tag{9}$$

For each pair, the same rotational behavior is used (e.g. an identity basis, a rotated eigenbasis, or implicit Newton-Shulz basis), but the elementwise normalizations are handled differently. Note that the Newton-Shulz operator of Muon is implicitly a signed descent method, as it approximates the orthogonalization of $\bar{g}$ such that all singular values are $\pm 1$.

We find that utilizing variance adaptation consistently achieves stronger results than otherwise. This trend remains consistent across all three optimizer pairs, as shwon in Figure 4, and the performance difference is nontrivial – for example, the difference between Muon and Adamuon is almost as large as the difference between Adam and Muon itself, indicating that variance adaptation is roughly as important as the spectral-normalizing aspect of matrix whitening.

Notably, variance adaptation is a natural consequence of the original whitening metric (Equation (2)), but theoretical equivalences between matrix-whitening methods and spectral descent (Bernstein & Newhouse, 2024) often rely on "disabling the accumulation" and treating all methods as signed descent (in a basis of choice), which may not be capturing the full picture. In fact, comparing Adam and Muon may be *understating* the gains from Newton-Schulz orthogonalization; a more fine-grained comparison would be Signum vs. Muon, or Adam vs. AdaMuon. We believe that proposed optimizers that focus solely on orthogonalizing updates (Ahn et al., 2025; Lau et al., 2025) will gain from re-implementing variance-adaptation in some form.

### 6.1 WHY DOES VARIANCE ADAPTATION STILL WORK WHEN DONE AFTER ORTHOGONALIZATION?

Interestingly, variance adaptation appears to provide a benefit regardless of the specific basis in which the adaptation is performed in. In SOAP, variance adaptation is performed in the *rotated eigenbasis*, as a pure alternative to signed descent. The same exchange is done in Adam versus Signum. However, in the AdaMuon setup, variance adaptation is performed in the *original elementwise basis*, after the update has already been spectrally-normalized via the Newton-Schulz iterations.

| Method | Val Loss | Walltime | Memory Usage |
|---|---|---|---|
| **Elementwise Basis** | | | |
| Sign [Signum] | $3.008$ $_{\pm.008}$ | 1.0 | $2n^2$ |
| Sign + Lookahead [Lion] | $3.008$ $_{\pm.008}$ | 1.0 | $2n^2$ |
| Variance-Full ($\beta_1 = \beta_2$) | $2.994$ $_{\pm.008}$ | 1.0 | $3n^2$ |
| **Variance-Factorized [Adafactor]** | **$2.989$** $_{\pm.008}$ | 1.0 | $2n^2 + 2n$ |
| **Variance-Full [Adam]** | **$2.982$** $_{\pm.008}$ | 1.0 | $3n^2$ |
| **Shampoo Basis (Every 100)** | | | |
| Sign [SPlus] | $2.961$ $_{\pm.003}$ | 1.2 | $4n^2$ |
| **Sign + Lookahead** | **$2.949$** $_{\pm.003}$ | 1.2 | $4n^2$ |
| Variance-Full ($\beta_1 = \beta_2$) | $2.952$ $_{\pm.003}$ | 1.2 | $5n^2$ |
| **Variance-Factorized** | **$2.946$** $_{\pm.003}$ | 1.2 | $4n^2 + 4n$ |
| **Variance-Full [SOAP]** | **$2.946$** $_{\pm.003}$ | 1.2 | $5n^2$ |
| **Newton-Schulz "Basis"** | | | |
| Sign | $2.964$ $_{\pm.003}$ | 1.07 | $2n^2$ |
| Sign + Lookahead [Muon] | $2.961$ $_{\pm.003}$ | 1.07 | $2n^2$ |
| Variance-Full ($\beta_1 = \beta_2$) | $2.953$ $_{\pm.003}$ | 1.07 | $3n^2$ |
| **Variance-Factorized** | **$2.943$** $_{\pm.003}$ | 1.07 | $2n^2 + 2n$ |
| Variance-Full [AdaMuon] | $2.950$ $_{\pm.003}$ | 1.07 | $3n^2$ |

Table 3: **Ablations on variance-adaptation across three optimizer families.** When a specific combination resembles a previously proposed method, we include that method in brackets.

In both cases, variance adaptation provides a reliable performance boost. One understanding that may explain the this phenomenon is the interpretation of variance adaptation as a heuristic for dynamically adjusting a trust region in proportion to a signal-to-noise ratio. As described in (Orvieto & Gower, 2025), when $\beta_1 = \beta_2$, Adam can be re-written as:

$$\textbf{Adam:} \quad \text{sign}(\bar{g}) \cdot \frac{1}{\sqrt{1 + \bar{\sigma}^2/\bar{g}^2}} \tag{10}$$

where $\bar{\sigma}^2 = \beta \cdot \text{EMA}_\beta \left[ (\bar{g} - g)^2 \right]$, i.e. an exponential moving average of the *centered* variance of gradients. Under this interpretation, the variance adaptation term serves as a dynamic learning-rate adjustment and does not necessarily have to share the same basis as the 'sign' term.

For this reason, we argue that matrix-whitening as described in Equation (2) serves *two* interpretable purposes. The first is to spectrally normalize updates, in effect reducing the learning rates of correlated parameters to prevent over-updating. The second is to further modulate these learning rates by a signal-to-noise term. While typically performed together, these two transformations can also be *decoupled* and done separately, as is the case in AdaMuon.

As a didactic example, we consider the "SPA" algorithm (SPlus-then-Adam), that first spectrally-normalizes updates with SPlus, and performs variance-modulation *afterwards*:

$$\textbf{SPA:} \; \text{unrot}(\text{sign}(\text{rot}(\bar{g}))) \; \oslash \; \mathbb{E}[\text{unrot}(\text{sign}(\text{rot}(\bar{g}))^2]^{-1/2} \tag{11}$$

When tuned, this addition achieves a final validation loss of 2.955, improving upon SPlus (albeit to a lesser degree than SOAP). We leave further examination on how to reconcile the proper bases for spectral-normalization and variance-adaptation to future investigation.

## 6.2 CAN LOOKAHEAD STRATEGIES REPLACE VARIANCE ADAPTATION?

The downside of variance adaptation is that one must keep track of an additional set of parameters in memory. Signed methods employing "lookahead" techniques (e.g. Lion Chen et al. (2023) and under loose interpretations MARS Yuan et al. (2024) or Cautious optimizers (Liang et al., 2024)) are a way to approximate this behavior without the additional memory cost. The general idea is to calculate the sign over $(1 - \beta_3)\bar{g} + \beta_3 g$, where $\beta_3$ is a new hyperparameter. For high-variance

gradients, the intuition is that the sign will flip more often between subsequent updates, resulting in a smaller overall change. This can also be interpreted as a generalization of Nesterov momentum (Dozat, 2016) which fixes $\beta_3 = 1 - \beta_1$. In Table 3, we sweep over $\beta_3$ for lookahead variants of the signed optimizers, and find that while gains can be achieved, these variants cannot reach the performance of variance-adapted variants (and require a sensitive additional hyperparameter).

### 6.3 CAN LOW-RANK FACTORIZATION REDUCE THE MEMORY FOOTPRINT OF VARIANCE ADAPTATION?

An alternate way to reduce memory requirements is to utilize a rank-1 approximation of the variance buffer, reducing memory usage from $mn$ to $m+n$. We find that factorized variance estimators retain almost exactly the same performance as the full matrices, and at times even improve performance. We utilize the following scaled Adafactor (Shazeer & Stern, 2018) update:

$$v_L \leftarrow (1 - \beta_2) \cdot v_L + (1 - \beta_2) \cdot \text{mean(G, axis=0)} \tag{12}$$

$$v_R \leftarrow (1 - \beta_2) \cdot v_R + (1 - \beta_2) \cdot \text{mean(G, axis=1)} \tag{13}$$

$$U = \bar{G} \oslash (v_L v_R^T) \cdot (\text{len}(v_R)/\text{sum}(v_L)) \tag{14}$$

As shown in Table 3 under the "Variance-Factored" label, using a rank-1 factorization results in negligible performance changes. For the specific case of Muon, the factorized variance estimator even *improves* performance over the full matrix estimator. We hypothesize that this may be due to a bias-variance tradeoff in the variance estimator. Taking the view from Section 6.1, variance adaptation can be seen as assigning a dynamic learning-rate to specific parameters, and this adaptivity may be more effective in practice if averaged over multiple parameters sharing a natural relationship, such as the input/output bases of a weight matrix (Morwani et al., 2024).

## 7 DISCUSSION AND CONCLUSION

In this work, we undertook a deconstruction of various matrix-whitening optimizers under a carefully-tuned experimental setup. Using this setting, we find evidence that matrix-whitening performs *two* key transformations–spectral normalization and variance adaptation. However, not all practical methods achieve both transformations. Spectrally-normalized methods outperform their elementwise counterparts, and variance-adapted methods outperform their sign-descent counterparts. Notably, these two components may be implemented in a *decoupled* manner, opening up the design space for future optimizers.

**Limitations.** We intentionally opt for a limited breadth of scope for our experiments – a single model architecture under a single language modelling objective – in favor of greater depth of ablations. While previous papers have also adopted a similar scope (Zhao et al., 2024; Liu et al., 2023), conclusions may transfer to varying degrees to different model sizes and data ratios, and we refer to (Wen et al., 2025) for a recent exploration of these scaling laws.

While we carefully sweep the learning rate, weight decay, $\beta_1$, and $\beta_2$, we did not search over other hyperparameters such as Adam's $\epsilon$, warmup, or alternate learning rate schedules. Wall-clock times are presented as a reference point, but wall-clock times may differ greatly depending on the specific hardware configuration used in training.

In this work, we focused on the core matrix-whitening behavior of optimizers. We intentionally did not consider changes to other aspects of optimization such as the initialization, momentum buffer, or automatic learning rate schedules, which may yield additional benefits.

**A challenge.** In our specific setting, the best methods improved over Adam by around $0.4$ final validation loss, moving from 2.98 to 2.94. The ablations we consider in the paper are able to effect changes on the order of $\pm 0.2$ validation loss. However, we believe that a more prominent jump in performance will require additional significant insights, perhaps beyond the abstraction of matrix-whitening. We present the challenge, *is there an alternate preconditioning scheme that can double these gains, and achieve a validation loss of $< 2.90$ under equivalent constraints?*

## REPRODUCIBILITY STATEMENT

We provide the exact code required to fully reproduce the results at https://anonymous.4open.science/r/matrix-whitening-submit-5DF2/, and describe the full experimental details in the Appendix.

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

## A APPENDIX

### A.1 DO CERTAIN PRECONDITIONING BASES MATTER MORE THAN OTHERS?

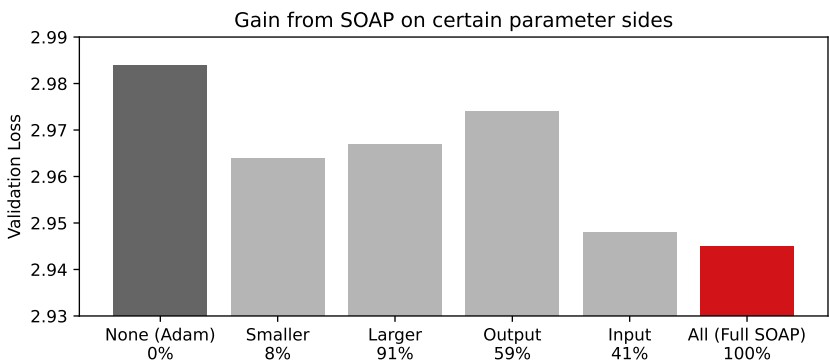

Figure 5: SOAP-100, with matrices preconditioned using only one side.

In the standard Shampoo (and SOAP) formulation, preconditioners are learned for both the input and output bases of each dense matrix. It has been suggested in previous works (Vyas et al., 2024; Bernstein & Newhouse, 2024; Vyas et al.) that only one of these preconditioners may be needed. Indeed, the orthogonalization view of the Shampoo update allows us to equate:

$$(GG^T)^{-1/4} \, G \, (G^TG)^{-1/4} \quad = \quad (GG^T)^{-1/2}G \quad = \quad G(G^TG)^{-1/2} \quad = \quad UV^T,$$

and we can take the matrix inverse for the smaller dimension. However, this equivalence does not hold in general when the left/right preconditioners are estimated via *historical* gradients.

In Figure 5, we compare a variant of SOAP where only half of the preconditioning matrices are used. We consider four strategies 1) the smaller dimension, 2) the larger dimension, 3) the input bases to each dense layer, and 4) the output basis. We find that **preconditioning the input basis recovers a majority of the performance of full SOAP.** This strategy reduces the memory overhead of SOAP to around 41% of the original. Notably, input-basis preconditioning outperforms using the larger dimension, implying an asymmetry.

### A.2 DO CERTAIN PARAMETER SUBSETS MATTER MORE THAN OTHERS?

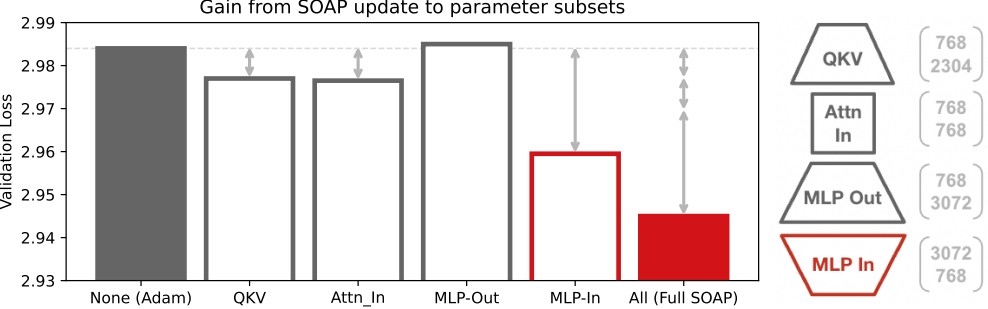

Figure 6: **Gains from using SOAP vs. Adam are roughly additive among independent parameter groups.** Each bar represents a Transformer trained with SOAP-100 applied to only that parameter type, and Adam applied otherwise. Learning rates are tuned, other hyperparameters are inherited from the baselines. Notably, a large percentage of the performance gain comes from preconditioning the **MLP-In** parameters, implying that the incoming 3072-length activations may be highly correlated.

A natural question is to ask whether it is important that all dense parameters are preconditioned via matrix-whitening methods, or if certain parameters types play an outsized impact on performance. In Figure 6, we examine this question by applying SOAP updates only to specific subsets of parameters, and using a default Adam optimizer otherwise. Learning rates are tuned independently per setting.

Notably, we find that the "MLP In" matrix benefits the most from preconditioning. One hypothesis for this finding is that the input features to this dense layer, a 3072-dimensional vector modulated by the 'gelu' activation, are highly linearly correlated. This should not be a surprising claim, as these features are the result of a linear projection from the 768-dimensional residual stream and thus are rank-deficient before the 'gelu' activation. If this is true, a naive Adam update may result in the simultaneous change of many parameters that affect the same singular vector in the update matrix, resulting in a greater-than-desired change. In contrast, a matrix-whitening optimizer can properly normalize to prevent this behavior.

An additional observation is that the gain in performance from matrix-whitening various parameters are roughly additive. This may imply that utilizing a matrix-whitening optimizer is helpful in so far as taking the appropriate update to prevent overshooting, but does not lead to branching changes across other parameters.

## A.3 ADAM STEPS

When estimating the number of steps that Adam would have taken to reach an equivalent validation loss, as reported in Table 2, we report the two ranges as $10000/T$, where $T$ is the minimum number of steps that Adam must be run to achieve an equivalent validation loss. We search over a resolution of $\pm 500$ steps, and report the lower and upper bins. Note that these are independent runs, as we utilize a cosine learning rate schedule, and we sweep over learning rate independently.

We provide this metric to avoid a potentially misleading alternative of "steps to reach Adam *within* a training run", as certain optimizers may reach lower validation losses faster, but fail near the end of training (Wen et al., 2025). Instead, we only consider the final validation losses of any experiment, and compare against final Adam validation losses.

Table 4: **Final validation loss achieved by Adam, under additional training steps.**

| Max Steps | Validation Loss |
|-----------|-----------------|
| 10000 | 2.984 |
| 11500 | 2.967 |
| 12000 | 2.964 |
| 12500 | 2.955 |
| 13000 | 2.952 |
| 13500 | 2.947 |
| 14000 | 2.944 |
| 14500 | 2.940 |
| 15000 | 2.938 |

## A.4 HYPERPARAMETERS IN EXPERIMENTS

In the following tables, we log the trials used throughout the paper to sweep over hyperparameters. Our criteria is that within the desired resolution, all four hyperparameters should be at their respective local optimums, as shown in Figure 2. For Shampoo-10, SPlus-10, and SOAP-10, we could not conduct an exhaustive search due to computational requirements, and we instead present a best-effort setting.

Table 5: **Shared hyperparameters across model training.**

| Hyperparameter | Value |
| --- | --- |
| Adam LR (Embed) | 0.01 |
| Adam LR (Output Head) | 0.01 |
| Adam LR (Layernorm) | 0.01 |
| Weight Decay (Embed) | 0 |
| Weight Decay (Output Head) | 0.001 |
| Weight Decay (Layernorm) | 0 |
| Adam $\beta_1$ (Embed/Output/Layernorm) | 0.9 |
| Adam $\beta_2$ (Embed/Output/Layernorm) | 0.99 |
| LR Warmup | 200 steps |
| LR Decay | Cosine |
| Sequence Length | 256 |
| Batch Size | 1024 |
| Training Iterations | $10,000$ |
| Weights Precision | fp32 |
| Optimizer Precision | fp32 |
| Activation Precision | bf16 |
| Hidden Size | 768 |
| MLP Ratio | 4 |
| Attention Heads | 12 |
| Num Blocks | 12 |

Table 6: **Hyperparameter sweeps for Adam.**

| Learning Rate | Weight Decay | $\beta_1$ | $\beta_2$ | Valid Loss |
|---|---|---|---|---|
| 0.01 | 1.0 | 0.95 | 0.99 | 4.564 |
| 0.001 | 10 | 0.95 | 0.99 | 3.366 |
| 0.00316 | 1.0 | 0.9 | 0.99 | 3.255 |
| 0.0001 | 1.0 | 0.95 | 0.99 | 3.196 |
| 0.001 | 3.162 | 0.9 | 0.99 | 3.182 |
| 0.00316 | 1.0 | 0.95 | 0.99 | 3.072 |
| 0.001 | 1.0 | 0.9 | 0.997 | 3.071 |
| 0.001 | 1.0 | 0.684 | 0.99 | 3.062 |
| 0.00075 | 1.0 | 0.677 | 0.99 | 3.057 |
| 0.001 | 3.162 | 0.968 | 0.99 | 3.039 |
| 0.001 | 3.16 | 0.95 | 0.99 | 3.035 |
| 0.001 | 1.0 | 0.99 | 0.99 | 3.025 |
| 0.00032 | 1.0 | 0.9 | 0.99 | 3.022 |
| 0.00032 | 1.0 | 0.95 | 0.99 | 3.020 |
| 0.00075 | 3.162 | 0.898 | 0.99 | 3.018 |
| 0.001 | 1.0 | 0.842 | 0.99 | 3.011 |
| 0.001 | 1.778 | 0.9 | 0.99 | 3.011 |
| 0.00178 | 1.0 | 0.968 | 0.99 | 3.006 |
| 0.00178 | 1.0 | 0.968 | 0.99 | 3.006 |
| 0.001 | 1.0 | 0.984 | 0.99 | 3.006 |
| 0.00042 | 1.0 | 0.898 | 0.99 | 3.006 |
| 0.001 | 0.1 | 0.95 | 0.99 | 3.005 |
| 0.001 | 0.316 | 0.968 | 0.99 | 3.004 |
| 0.001 | 1.778 | 0.968 | 0.99 | 3.004 |
| 0.00178 | 1.0 | 0.9 | 0.99 | 3.003 |
| 0.001 | 1.0 | 0.822 | 0.99 | 3.003 |
| 0.00075 | 0.316 | 0.898 | 0.99 | 3.000 |
| 0.00133 | 1.0 | 0.898 | 0.99 | 2.997 |
| 0.00133 | 1.0 | 0.968 | 0.99 | 2.996 |
| 0.00075 | 1.778 | 0.898 | 0.99 | 2.995 |
| 0.001 | 1.0 | 0.968 | 0.99 | 2.994 |
| 0.00056 | 1.0 | 0.9 | 0.99 | 2.994 |
| 0.00056 | 1.0 | 0.898 | 0.99 | 2.994 |
| 0.001 | 0.316 | 0.9 | 0.99 | 2.992 |
| 0.00056 | 1.0 | 0.968 | 0.99 | 2.992 |
| 0.001 | 0.316 | 0.95 | 0.99 | 2.991 |
| 0.00133 | 1.0 | 0.9 | 0.99 | 2.991 |
| 0.001 | 1.0 | 0.968 | 0.99 | 2.991 |
| 0.001 | 1.0 | 0.9 | 0.968 | 2.989 |
| 0.001 | 1.0 | 0.899 | 0.99 | 2.988 |
| 0.00075 | 1.0 | 0.968 | 0.99 | 2.988 |
| 0.00075 | 1.0 | 0.898 | 0.968 | 2.988 |
| 0.00075 | 1.0 | 0.898 | 0.997 | 2.987 |
| 0.00075 | 1.0 | 0.9 | 0.99 | 2.986 |
| 0.00075 | 1.0 | 0.898 | 0.99 | 2.985 |
| 0.00075 | 1.0 | 0.968 | 0.99 | 2.985 |
| 0.001 | 0.562 | 0.9 | 0.99 | 2.984 |
| 0.001 | 1.0 | 0.95 | 0.99 | 2.983 |
| 0.001 | 1.0 | 0.9 | 0.99 | 2.982 |
| 0.001 | 1.0 | 0.898 | 0.99 | 2.982 |
| 0.001 | 1.0 | 0.944 | 0.99 | 2.982 |

Table 7: **Hyperparameter sweeps for Signum.**

| Learning Rate | Weight Decay | $\beta_1$ | $\beta_2$ | Valid Loss |
|---|---|---|---|---|
| 0.00032 | 1.0 | 0.968 | 0 | 6.624 |
| 0.00056 | 1.0 | 0.9 | 0 | 6.496 |
| 0.00032 | 3.162 | 0.968 | 0 | 6.313 |
| 0.00056 | 3.162 | 0.9 | 0 | 4.121 |
| 0.00042 | 1.0 | 0.9 | 0 | 3.160 |
| 0.00042 | 3.162 | 0.9 | 0 | 3.136 |
| 0.00032 | 0.316 | 0.9 | 0 | 3.076 |
| 0.00032 | 0.562 | 0.9 | 0 | 3.065 |
| 0.00018 | 3.162 | 0.968 | 0 | 3.060 |
| 0.00032 | 1.0 | 0.9 | 0 | 3.056 |
| 0.00032 | 9.998 | 0.9 | 0 | 3.055 |
| 0.00032 | 1.0 | 0.9 | 0 | 3.051 |
| 0.00032 | 5.622 | 0.9 | 0 | 3.045 |
| 0.00032 | 1.0 | 0.684 | 0 | 3.044 |
| 0.00024 | 1.0 | 0.9 | 0 | 3.043 |
| 0.00032 | 1.778 | 0.9 | 0 | 3.039 |
| 0.00032 | 1.778 | 0.9 | 0 | 3.038 |
| 0.00018 | 3.162 | 0.684 | 0 | 3.033 |
| 0.00018 | 1.0 | 0.9 | 0 | 3.032 |
| 0.00032 | 3.162 | 0.684 | 0 | 3.031 |
| 0.00031 | 3.162 | 0.9 | 0 | 3.030 |
| 0.00018 | 1.0 | 0.9 | 0 | 3.029 |
| 0.00032 | 3.162 | 0.9 | 0 | 3.028 |
| 0.00032 | 3.162 | 0.9 | 0 | 3.025 |
| 0.00024 | 3.162 | 0.9 | 0 | 3.022 |
| 0.00018 | 9.998 | 0.9 | 0 | 3.020 |
| 0.0001 | 3.162 | 0.9 | 0 | 3.020 |
| 0.00018 | 1.778 | 0.9 | 0 | 3.019 |
| 0.00024 | 3.162 | 0.9 | 0 | 3.017 |
| 0.00018 | 3.162 | 0.9 | 0 | 3.008 |
| 0.00013 | 3.162 | 0.9 | 0 | 3.007 |
| 0.00018 | 5.622 | 0.9 | 0 | 3.006 |

Table 8: **Hyperparameter sweeps for PSGD.**

| Learning Rate | Weight Decay | $\beta_1$ | $\beta_2$ | Valid Loss |
|---|---|---|---|---|
| 0.00035 | 0.001 | 0.968 | 0 | 3.666 |
| 0.0002 | 0.001 | 0.968 | 0 | 3.456 |
| 0.00083 | 0.001 | 0.968 | 0 | 2.996 |
| 8e-05 | 0.001 | 0.968 | 0 | 2.994 |
| 0.00026 | 0.001 | 0.99 | 0 | 2.990 |
| 0.00026 | 0.001 | 0.943 | 0 | 2.990 |
| 0.00026 | 0.001 | 0.899 | 0 | 2.988 |
| 0.00015 | 0.001 | 0.968 | 0 | 2.983 |
| 0.00047 | 0.001 | 0.968 | 0 | 2.982 |
| 0.00026 | 0.001 | 0.982 | 0 | 2.977 |
| 0.0002 | 0.001 | 0.968 | 0 | 2.977 |
| 0.00026 | 0.0 | 0.968 | 0 | 2.976 |
| 0.00035 | 0.001 | 0.968 | 0 | 2.975 |
| 0.00026 | 0.003 | 0.968 | 0 | 2.975 |
| 0.00026 | 0.002 | 0.968 | 0 | 2.974 |
| 0.00026 | 0.001 | 0.968 | 0 | 2.973 |

Table 9: **Hyperparameter sweeps for SPlus-100.**

| Learning Rate | Weight Decay | $\beta_1$ | $\beta_2$ | Valid Loss |
|---|---|---|---|---|
| 0.1 | 0.032 | 0.968 | 0.99 | 3.434 |
| 0.1 | 0.018 | 0.968 | 0.99 | 3.226 |
| 0.1 | 0.01 | 0.997 | 0.968 | 3.161 |
| 0.1778 | 0.01 | 0.968 | 0.99 | 3.120 |
| 0.1333 | 0.01 | 0.968 | 0.99 | 3.100 |
| 0.03163 | 0.01 | 0.99 | 0.968 | 3.004 |
| 0.1 | 0.032 | 0.99 | 0.99 | 2.999 |
| 0.3162 | 0.01 | 0.99 | 0.968 | 2.996 |
| 0.1 | 0.032 | 0.99 | 0.968 | 2.995 |
| 0.1 | 0.003 | 0.968 | 0.99 | 2.985 |
| 0.05624 | 0.01 | 0.968 | 0.99 | 2.983 |
| 0.1 | 0.018 | 0.968 | 0.99 | 2.981 |
| 0.1 | 0.01 | 0.899 | 0.99 | 2.981 |
| 0.1 | 0.01 | 0.997 | 0.99 | 2.978 |
| 0.1 | 0.01 | 0.997 | 0.99 | 2.977 |
| 0.1 | 0.003 | 0.99 | 0.968 | 2.976 |
| 0.1778 | 0.01 | 0.99 | 0.99 | 2.975 |
| 0.1 | 0.003 | 0.99 | 0.99 | 2.974 |
| 0.1778 | 0.01 | 0.99 | 0.968 | 2.973 |
| 0.07502 | 0.01 | 0.968 | 0.99 | 2.973 |
| 0.05624 | 0.01 | 0.99 | 0.99 | 2.972 |
| 0.1 | 0.018 | 0.99 | 0.968 | 2.972 |
| 0.1 | 0.018 | 0.99 | 0.99 | 2.972 |
| 0.05624 | 0.01 | 0.99 | 0.968 | 2.971 |
| 0.1 | 0.01 | 0.968 | 0.99 | 2.971 |
| 0.1 | 0.01 | 0.968 | 0.99 | 2.970 |
| 0.1 | 0.01 | 0.968 | 0.997 | 2.969 |
| 0.1 | 0.01 | 0.968 | 0.968 | 2.969 |
| 0.07502 | 0.01 | 0.99 | 0.968 | 2.969 |
| 0.1333 | 0.01 | 0.99 | 0.99 | 2.969 |
| 0.1 | 0.01 | 0.994 | 0.968 | 2.969 |
| 0.1333 | 0.01 | 0.99 | 0.99 | 2.968 |
| 0.1 | 0.006 | 0.99 | 0.968 | 2.967 |
| 0.1 | 0.01 | 0.99 | 0.899 | 2.966 |
| 0.1 | 0.01 | 0.982 | 0.968 | 2.966 |
| 0.07502 | 0.01 | 0.99 | 0.99 | 2.965 |
| 0.1 | 0.01 | 0.99 | 0.997 | 2.965 |
| 0.1 | 0.01 | 0.99 | 0.99 | 2.965 |
| 0.1333 | 0.01 | 0.99 | 0.968 | 2.965 |
| 0.1 | 0.01 | 0.99 | 0.997 | 2.964 |
| 0.1 | 0.01 | 0.99 | 0.99 | 2.964 |
| 0.1 | 0.01 | 0.99 | 0.99 | 2.963 |
| 0.1 | 0.01 | 0.99 | 0.968 | 2.963 |
| 0.1 | 0.01 | 0.99 | 0.968 | 2.962 |

Table 10: **Hyperparameter sweeps for SOAP-100.**

| Learning Rate | Weight Decay | $\beta_1$ | $\beta_2$ | Valid Loss |
|---|---|---|---|---|
| 0.00132 | 3.162 | 0.968 | 0.99 | 3.020 |
| 0.00132 | 3.162 | 0.968 | 0.99 | 3.008 |
| 0.00175 | 0.316 | 0.822 | 0.99 | 2.980 |
| 0.00235 | 1.0 | 0.968 | 0.99 | 2.980 |
| 0.00055 | 0.316 | 0.9 | 0.99 | 2.975 |
| 0.00132 | 1.778 | 0.968 | 0.99 | 2.969 |
| 0.00132 | 1.778 | 0.968 | 0.99 | 2.968 |
| 0.00235 | 1.0 | 0.968 | 0.99 | 2.967 |
| 0.00074 | 0.316 | 0.9 | 0.99 | 2.965 |
| 0.00098 | 0.316 | 0.9 | 0.99 | 2.958 |
| 0.00132 | 0.1 | 0.9 | 0.99 | 2.958 |
| 0.00132 | 1.0 | 0.899 | 0.99 | 2.958 |
| 0.00074 | 1.0 | 0.968 | 0.99 | 2.957 |
| 0.00176 | 1.0 | 0.968 | 0.99 | 2.957 |
| 0.00132 | 1.0 | 0.968 | 0.997 | 2.957 |
| 0.00099 | 0.316 | 0.9 | 0.99 | 2.956 |
| 0.00132 | 1.0 | 0.99 | 0.99 | 2.955 |
| 0.00099 | 1.0 | 0.968 | 0.99 | 2.955 |
| 0.00132 | 1.0 | 0.99 | 0.99 | 2.954 |
| 0.00132 | 1.0 | 0.968 | 0.968 | 2.954 |
| 0.00132 | 0.316 | 0.968 | 0.99 | 2.953 |
| 0.00233 | 0.316 | 0.9 | 0.99 | 2.953 |
| 0.00132 | 1.0 | 0.968 | 0.99 | 2.953 |
| 0.00132 | 0.316 | 0.9 | 0.968 | 2.952 |
| 0.00132 | 1.0 | 0.968 | 0.99 | 2.952 |
| 0.00132 | 0.316 | 0.968 | 0.99 | 2.952 |
| 0.00132 | 0.316 | 0.9 | 0.99 | 2.952 |
| 0.00132 | 1.0 | 0.899 | 0.99 | 2.952 |
| 0.00175 | 0.561 | 0.9 | 0.99 | 2.952 |
| 0.00175 | 0.178 | 0.9 | 0.99 | 2.951 |
| 0.00132 | 0.999 | 0.9 | 0.99 | 2.951 |
| 0.00099 | 1.0 | 0.968 | 0.99 | 2.951 |
| 0.00132 | 0.562 | 0.9 | 0.99 | 2.951 |
| 0.00175 | 0.177 | 0.9 | 0.99 | 2.951 |
| 0.00175 | 0.561 | 0.9 | 0.968 | 2.950 |
| 0.00131 | 0.316 | 0.9 | 0.99 | 2.950 |
| 0.00132 | 0.316 | 0.9 | 0.997 | 2.950 |
| 0.00131 | 0.561 | 0.9 | 0.99 | 2.950 |
| 0.00175 | 0.316 | 0.9 | 0.968 | 2.950 |
| 0.00098 | 0.561 | 0.9 | 0.99 | 2.950 |
| 0.00175 | 0.561 | 0.968 | 0.99 | 2.950 |
| 0.00175 | 0.316 | 0.944 | 0.99 | 2.949 |
| 0.00235 | 0.316 | 0.9 | 0.99 | 2.949 |
| 0.00175 | 0.561 | 0.9 | 0.997 | 2.948 |
| 0.00175 | 0.316 | 0.9 | 0.997 | 2.948 |
| 0.00175 | 0.316 | 0.9 | 0.99 | 2.946 |

Table 11: **Hyperparameter sweeps for SOAP-10.**

| Learning Rate | Weight Decay | $\beta_1$ | $\beta_2$ | Valid Loss |
|---|---|---|---|---|
| 0.00175 | 0.316 | 0.684 | 0.99 | 3.119 |
| 0.00175 | 0.316 | 0.968 | 0.968 | 3.041 |
| 0.00175 | 0.316 | 0.899 | 0.99 | 3.033 |
| 0.00175 | 0.316 | 0.968 | 0.997 | 3.028 |
| 0.00175 | 0.999 | 0.9 | 0.99 | 2.997 |
| 0.00055 | 0.316 | 0.968 | 0.99 | 2.972 |
| 0.00074 | 0.316 | 0.968 | 0.99 | 2.964 |
| 0.00175 | 0.1 | 0.9 | 0.99 | 2.962 |
| 0.00132 | 0.1 | 0.968 | 0.99 | 2.962 |
| 0.00098 | 0.316 | 0.9 | 0.99 | 2.958 |
| 0.00099 | 0.316 | 0.968 | 0.99 | 2.956 |
| 0.00175 | 0.178 | 0.9 | 0.99 | 2.955 |
| 0.00098 | 0.316 | 0.968 | 0.99 | 2.954 |
| 0.00175 | 0.562 | 0.9 | 0.99 | 2.954 |
| 0.00311 | 0.316 | 0.9 | 0.99 | 2.954 |
| 0.00175 | 0.316 | 0.9 | 0.997 | 2.953 |
| 0.00132 | 0.178 | 0.968 | 0.99 | 2.953 |
| 0.00175 | 0.1 | 0.968 | 0.99 | 2.952 |
| 0.00132 | 0.316 | 0.968 | 0.997 | 2.952 |
| 0.00175 | 0.999 | 0.968 | 0.99 | 2.952 |
| 0.00132 | 0.316 | 0.899 | 0.99 | 2.952 |
| 0.00132 | 0.316 | 0.99 | 0.99 | 2.951 |
| 0.00175 | 0.316 | 0.9 | 0.968 | 2.951 |
| 0.00131 | 0.316 | 0.9 | 0.99 | 2.949 |
| 0.00132 | 0.316 | 0.968 | 0.968 | 2.949 |
| 0.00553 | 0.316 | 0.968 | 0.99 | 2.948 |
| 0.00132 | 0.316 | 0.9 | 0.99 | 2.948 |
| 0.00132 | 0.999 | 0.968 | 0.99 | 2.947 |
| 0.00132 | 0.316 | 0.968 | 0.99 | 2.947 |
| 0.00131 | 0.316 | 0.968 | 0.99 | 2.947 |
| 0.00175 | 0.178 | 0.968 | 0.99 | 2.947 |
| 0.00233 | 0.316 | 0.9 | 0.99 | 2.947 |
| 0.00175 | 0.316 | 0.99 | 0.99 | 2.945 |
| 0.00175 | 0.316 | 0.9 | 0.99 | 2.945 |
| 0.00132 | 0.562 | 0.968 | 0.99 | 2.945 |
| 0.00175 | 0.316 | 0.9 | 0.99 | 2.944 |
| 0.00175 | 0.316 | 0.943 | 0.99 | 2.943 |
| 0.00175 | 0.562 | 0.968 | 0.99 | 2.943 |
| 0.00175 | 0.316 | 0.982 | 0.99 | 2.942 |
| 0.00175 | 0.316 | 0.968 | 0.99 | 2.942 |
| 0.00176 | 0.316 | 0.968 | 0.99 | 2.942 |
| 0.00235 | 0.316 | 0.968 | 0.99 | 2.941 |
| 0.00233 | 0.316 | 0.968 | 0.99 | 2.940 |
| 0.00311 | 0.316 | 0.968 | 0.99 | 2.939 |

Table 12: **Hyperparameter sweeps for Muon.**

| Learning Rate | Weight Decay | $\beta_1$ | $\beta_2$ | Valid Loss |
|---|---|---|---|---|
| 0.0578 | 0.1 | 0.95 | 0 | 3.194 |
| 0.00058 | 0.1 | 0.95 | 0 | 3.122 |
| 0.00578 | 1.0 | 0.95 | 0 | 3.040 |
| 0.00183 | 0.1 | 0.95 | 0 | 3.003 |
| 0.00244 | 0.1 | 0.9 | 0 | 2.998 |
| 0.00578 | 0.01 | 0.95 | 0 | 2.989 |
| 0.0077 | 0.316 | 0.9 | 0 | 2.986 |
| 0.02436 | 0.1 | 0.9 | 0 | 2.985 |
| 0.0077 | 0.316 | 0.968 | 0 | 2.985 |
| 0.0077 | 0.1 | 0.684 | 0 | 2.980 |
| 0.00578 | 0.032 | 0.95 | 0 | 2.980 |
| 0.00578 | 0.032 | 0.968 | 0 | 2.979 |
| 0.0077 | 0.032 | 0.968 | 0 | 2.978 |
| 0.01826 | 0.1 | 0.95 | 0 | 2.978 |
| 0.0077 | 0.032 | 0.9 | 0 | 2.978 |
| 0.00578 | 0.316 | 0.968 | 0 | 2.977 |
| 0.00578 | 0.316 | 0.95 | 0 | 2.976 |
| 0.0077 | 0.1 | 0.99 | 0 | 2.976 |
| 0.00433 | 0.1 | 0.9 | 0 | 2.975 |
| 0.00325 | 0.1 | 0.968 | 0 | 2.975 |
| 0.00578 | 0.1 | 0.99 | 0 | 2.974 |
| 0.0137 | 0.1 | 0.968 | 0 | 2.972 |
| 0.00578 | 0.1 | 0.984 | 0 | 2.970 |
| 0.00433 | 0.1 | 0.968 | 0 | 2.970 |
| 0.0077 | 0.178 | 0.968 | 0 | 2.969 |
| 0.0137 | 0.1 | 0.9 | 0 | 2.969 |
| 0.0077 | 0.1 | 0.822 | 0 | 2.969 |
| 0.00434 | 0.1 | 0.968 | 0 | 2.968 |
| 0.01028 | 0.1 | 0.968 | 0 | 2.968 |
| 0.01027 | 0.1 | 0.968 | 0 | 2.968 |
| 0.0077 | 0.056 | 0.9 | 0 | 2.968 |
| 0.0077 | 0.178 | 0.9 | 0 | 2.968 |
| 0.00578 | 0.1 | 0.9 | 0 | 2.968 |
| 0.00578 | 0.1 | 0.899 | 0 | 2.967 |
| 0.00578 | 0.178 | 0.968 | 0 | 2.967 |
| 0.00578 | 0.1 | 0.968 | 0 | 2.967 |
| 0.0077 | 0.1 | 0.968 | 0 | 2.967 |
| 0.0077 | 0.1 | 0.968 | 0 | 2.966 |
| 0.00578 | 0.1 | 0.95 | 0 | 2.966 |
| 0.00578 | 0.1 | 0.968 | 0 | 2.966 |
| 0.0077 | 0.1 | 0.944 | 0 | 2.965 |
| 0.01027 | 0.1 | 0.9 | 0 | 2.965 |
| 0.0077 | 0.1 | 0.9 | 0 | 2.965 |
| 0.0077 | 0.1 | 0.899 | 0 | 2.964 |

Table 13: **Hyperparameter sweeps for AdaMuon.**

| Learning Rate | Weight Decay | $\beta_1$ | $\beta_2$ | Valid Loss |
|---|---|---|---|---|
| 0.00042 | 0.1 | 0.968 | 0.99 | 2.987 |
| 0.00013 | 0.316 | 0.968 | 0.99 | 2.983 |
| 0.00056 | 0.316 | 0.968 | 0.99 | 2.983 |
| 0.00031 | 1.0 | 0.99 | 0.99 | 2.982 |
| 0.00031 | 9.998 | 0.9 | 0.99 | 2.982 |
| 0.00042 | 0.316 | 0.968 | 0.99 | 2.979 |
| 0.00042 | 0.316 | 0.968 | 0.997 | 2.978 |
| 0.00031 | 9.998 | 0.968 | 0.99 | 2.977 |
| 0.00023 | 0.316 | 0.968 | 0.99 | 2.976 |
| 0.00031 | 0.316 | 0.968 | 0.99 | 2.976 |
| 0.00042 | 0.316 | 0.968 | 0.968 | 2.975 |
| 0.00031 | 0.316 | 0.968 | 0.99 | 2.975 |
| 0.00042 | 0.316 | 0.899 | 0.99 | 2.970 |
| 0.00042 | 0.562 | 0.968 | 0.99 | 2.969 |
| 0.00018 | 1.0 | 0.968 | 0.99 | 2.968 |
| 0.0001 | 3.162 | 0.968 | 0.99 | 2.967 |
| 0.00055 | 1.0 | 0.968 | 0.99 | 2.967 |
| 0.00031 | 1.0 | 0.968 | 0.99 | 2.964 |
| 0.00055 | 3.162 | 0.968 | 0.99 | 2.963 |
| 0.00031 | 5.622 | 0.968 | 0.99 | 2.963 |
| 0.00031 | 1.0 | 0.968 | 0.997 | 2.963 |
| 0.00031 | 3.162 | 0.684 | 0.99 | 2.962 |
| 0.00031 | 3.162 | 0.99 | 0.99 | 2.962 |
| 0.00031 | 1.0 | 0.968 | 0.968 | 2.962 |
| 0.00031 | 1.0 | 0.9 | 0.99 | 2.962 |
| 0.00031 | 1.0 | 0.899 | 0.99 | 2.961 |
| 0.00031 | 1.0 | 0.968 | 0.99 | 2.961 |
| 0.00031 | 5.622 | 0.9 | 0.99 | 2.961 |
| 0.00042 | 0.999 | 0.968 | 0.99 | 2.960 |
| 0.00018 | 3.162 | 0.9 | 0.99 | 2.960 |
| 0.00055 | 3.162 | 0.9 | 0.99 | 2.957 |
| 0.00031 | 3.162 | 0.982 | 0.99 | 2.957 |
| 0.00042 | 3.162 | 0.968 | 0.99 | 2.956 |
| 0.00031 | 1.778 | 0.968 | 0.99 | 2.956 |
| 0.00018 | 3.162 | 0.968 | 0.99 | 2.956 |
| 0.00031 | 1.778 | 0.968 | 0.99 | 2.955 |
| 0.00031 | 3.162 | 0.9 | 0.968 | 2.954 |
| 0.00031 | 3.162 | 0.9 | 0.99 | 2.954 |
| 0.00031 | 3.162 | 0.968 | 0.99 | 2.953 |
| 0.00031 | 3.162 | 0.968 | 0.968 | 2.953 |
| 0.00031 | 3.162 | 0.899 | 0.99 | 2.953 |
| 0.00023 | 3.162 | 0.9 | 0.99 | 2.952 |
| 0.00042 | 3.162 | 0.9 | 0.99 | 2.952 |
| 0.00031 | 3.162 | 0.968 | 0.997 | 2.952 |
| 0.00031 | 3.162 | 0.968 | 0.99 | 2.952 |
| 0.00023 | 3.162 | 0.968 | 0.99 | 2.952 |
| 0.00031 | 3.162 | 0.943 | 0.99 | 2.951 |
| 0.00031 | 3.162 | 0.968 | 0.99 | 2.950 |

