# OpenReview forum: "What really matters in matrix whitening optimizers?"
_ICLR.cc/2026/Conference — Submitted to ICLR 2026_

### Official Review · Reviewer_veBi · 2025-10-31

**Soundness:** 2
**Presentation:** 3
**Contribution:** 2
**Rating:** 4
**Confidence:** 2

**Summary:**

This paper systematically deconstructs and analyzes the factors contributing to the performance improvements of modern matrix-whitening optimizers such as Shampoo, SOAP, and Muon. The authors argue that the success of these optimizers is not solely attributed to 'spectral normalization', but that 'variance adaptation'—an often-overlooked component—is critically important. Experiments demonstrated that optimizers incorporating variance adaptation (Adam, SOAP, AdaMuon) consistently outperformed those without it (Signum, SPlus, Muon). This suggests that the two components of a matrix-whitening optimizer can be applied in a decoupled manner.

**Strengths:**

1.	The analytical approach itself, which deconstructs optimizers along the two axes of 'spectral normalization' and 'variance adaptation', is original.

2.	Although some experiments were 'best-effort' , the fact that the other version optimizers underwent thorough tuning of four key hyperparameters —which was transparently disclosed in the appendix—shows an effort to adhere to scientific procedures.

**Weaknesses:**

1.	Lack of generalization: All conclusions rely solely on a single model (GPT-2 Base) , a single task (Language Modeling) , and a single dataset (OpenWebText).

2.	Insufficient evaluation metrics: The sole measure of final performance is 'Validation Loss'. The paper fails to demonstrate whether faster convergence or a slight loss improvement translates to actual performance gains on downstream tasks.

3.	Baseline failure: The analysis completely lacks an explanation for why the key baseline, Shampoo-100, failed to converge, which lowers the experiment's reliability.

4.	Ambiguity: The use of a 'simplified' AdaMuon and the 'best-effort' tuning for N=10 versions are factors that compromise the fairness of the comparisons.

**Questions:**

1.	It would be difficult to generalize findings based on only a single model and dataset . Can you provide evidence that the paper's claims hold true for other transformer-based models and datasets?

2.	Can you present evidence that the marginal improvements in validation loss lead to statistically significant performance gains on practical downstream tasks for LLMs, such as the GLUE benchmark?

3.	Please provide a clear explanation for the convergence failure of Shampoo-100.

---

> ### Author Response · Authors · 2025-11-18
>
> Thank you for your review. We address your concerns below:
>
> **On generalization of findings and evaluation criteria**.
>
> This is a fair concern. We first note that many published prior works evaluate optimizers solely on the language modelling setting [1][2][3]. Our model/dataset is not arbitrary, but rather one that closely matches e.g. the nanogpt speedrun [4].  Additionally, among these and other past works (see Related Work para 3), validation loss is the standard metric for comparing optimizer performance.
>
> That said, we agree that the best way to concrerely show generalization is to run more experiments. We have replicated our study on a separate **diffusion modelling** objective, training a diffusion transformer on the task of latent diffusion on the Imagenet dataset. We utilize the stable diffusion VAE, a patch size of 2, and a flow-matching loss, following standard practice [5]. We confirm that our proposed trends generalize to this setting -- Spectral-normalized methods (SOAP/Muon families) outperform elementwise methods (Adam), and variance-adaptive variants (SOAP vs SPlus), (Adamuon vs Muon) are superior.
>
> | Optimizer | Val Loss (Language Modelling) | Val Loss (Diffusion) |
> |-----------|----------|----------|
> | Signum    | 3.006             | 0.7902   |
> | Adam      | **2.982**         | **0.7890**   |
> | SPlus-100 | 2.962             | 0.7839   |
> | SOAP-100  | **2.946**         | **0.7816**   |
> | Muon      | 2.964             | 0.7813   |
> | Adamuon   | **2.950**         | **0.7800**   |
>
> **Confusing Shampoo Baseline.**
>
> It is a fair point, and it's true that the presented Shampoo-100 comparison is confusing. It has been shown in [5] that Shampoo is unstable when cached for 100 iterations, even when extensively tuned -- this is a byproduct of the stale preconditioner. **We emphasize that Shampoo is not a main comparison, it is SOAP-100/SPlus-100 that are the relevant comparisons.**  (Note prior works such as [2], [7], which do not even attempt a Shampoo comparison, and only use SOAP which is known to be more performant and stable). We included Shampoo (as well as the "best-effort" SOAP-10/SPlus-10) as a point of reference -- but these are not the main baselines, and we admit this is confusing for readers, and we will move these studies to the Appendix in a revised version. **Figures 3+4 and Table 3 compare Adam, Muon, and SOAP-100, all of which are tuned extensively**. We will clarify this.
>
> With regards to "Adamuon", we note that the Adamuon paper itself is concurrent to this work (and under review to this same ICLR conference). We call our variance-adapted method "Adamuon" to avoid introducing another name to a similar optimizer (see Table 3, where we have tried to match specific combinations to prior works when they are similar). However, the "simplified" version we consider in the paper is **precisely the minimal ablation of Muon to introduce variance-adaptation**. We are not aiming to compare distinct optimizers, but rather the individual components that build an optimizer recipe, which we believe will lead to cleaner scientific findings.
>
> We hope that the two points have addressed your concerns, and better explain the paper. Please let us know if any confusion remains.
>
> [1] Zhao, Rosie, Depen Morwani, David Brandfonbrener, Nikhil Vyas, and Sham Kakade. "Deconstructing what makes a good optimizer for language models." arXiv preprint arXiv:2407.07972 (2024).
>
> [2] Wen, Kaiyue, David Hall, Tengyu Ma, and Percy Liang. "Fantastic pretraining optimizers and where to find them." arXiv preprint arXiv:2509.02046 (2025).
>
> [3] H. Liu, Z. Li, D. Hall, P. Liang, and T. Ma. Sophia: A scalable stochastic second-order optimizer for language
> model pre-training, 2024a
>
> [4] https://github.com/KellerJordan/modded-nanogpt
>
> [5] Frans, Kevin, Sergey Levine, and Pieter Abbeel. "A Stable Whitening Optimizer for Efficient Neural Network Training." arXiv preprint arXiv:2506.07254 (2025).
>
> [6] Ma, Nanye, Mark Goldstein, Michael S. Albergo, Nicholas M. Boffi, Eric Vanden-Eijnden, and Saining Xie. "Sit: Exploring flow and diffusion-based generative models with scalable interpolant transformers." In European Conference on Computer Vision, pp. 23-40. Cham: Springer Nature Switzerland, 2024.
>
> [7] Vyas, Nikhil, Rosie Zhao, Depen Morwani, Mujin Kwun, and Sham Kakade. Improving SOAP using iterative whitening and Muon. 2025.

---

### Official Review · Reviewer_HjvJ · 2025-10-31

**Soundness:** 2
**Presentation:** 3
**Contribution:** 2
**Rating:** 2
**Confidence:** 3

**Summary:**

This study compares various optimizers based on the *whitening metric* by reporting the minimal validation loss achieved after training a GPT-2 transformer for next-token prediction on the OpenWebText dataset. The training consists of 10,000 updates with cosine learning rates following an initial warmup. The hyperparameter sweep includes four parameters: learning rate, weight decay, momentum EMA coefficient, and variance EMA coefficient. Other hyperparameters, such as Adam's epsilon (also used in other optimizers for regularizing inverses), are held constant.

The first key result confirms that all carefully tuned optimizers employing update rotation outperform the diagonal Adam optimizer.

Additionally, the study argues that Shampoo-style optimizers facilitate more accurate estimation of steepest spectral descent, based on a metric that I was not able to fully understand. Importantly, this improved spectral descent is not the sole factor behind the enhanced performance: a key takeaway is that, when comparing "variance-adaptation" and "sign" versions of equivalent optimizers side by side, the variance-adaptation variants consistently perform better.

**Strengths:**

The empirical comparison is of interest by itself, even though I think a much greater effort should be pursued across the whole community to provide larger scale controlled benchmarks.

The paper is careful not to overstate its findings. The limitations are clearly acknowledged, emphasizing that the results are based on a single experimental setup.

**Weaknesses:**

Surveying different optimization methods is inherently challenging due to the large number of hyperparameters, the multitude of optimizers proposed by the community, and the variety of available datasets—even for the same task of next-token prediction. While the current work is undoubtedly of interest, I am uncertain whether its findings are sufficiently general to warrant publication at ICLR. For example, using a different learning rate scheduler (such as cosine annealing) might lead to different conclusions.

Additionally, some design choices could be questioned. For instance, I believe that Adam's epsilon, which is arguably the second most important hyperparameter after the learning rate, should have been included in the hyperparameter sweep.

Another point of concern is that it is not clear what is being plotted in the left panel of Figure 3, nor is the corresponding discussion sufficiently explanatory.

**Questions:**

1. A clarification question: what is the "singular value of updates" (l.305, plotted in figure 3 ?), where the "update" is just a step in parameter space, i.e. a vector of all parameters of the LLM, for which I don't get the concept of "singular value".
2. A question on the relevant of publishing such work to ICLR: Do you think your findings would generalize to other setups, or are they specific to this architecture/dataset/training setup ? If so, how can you convince your readers ?

Then, I have some more minor questions/comments:

3. How are the hyperparameters of the "nonstandard" Adam optimizer (line 199) chosen?
4. It would be helpful to include a summary table of the update rules for all the benchmarked alternatives (lines 212–262).
5. Why limit the comparison to optimizers based on \sqrt{var} and sign only? Since implementing natural gradient-like optimizers (such as KFAC/EKFAC) involves not much additional effort, including them could provide a more theoretically grounded comparison. Do you have any thoughts on this?
6. Could you clarify whether you are using weight decay or L2 regularization? This distinction might matter, as discussed in "Three Mechanisms of Weight Decay Regularization" (ICLR 2019).
7. What about the training loss? Isn't that the quantity that the optimizers are directly minimizing?

---

> ### Author Response · Authors · 2025-11-18
>
> Thank you for your detailed comments. Please see our responses to your concerns below.
>
> **On the generality of findings**.
>
> You note you are "uncertain whether its findings are sufficiently general ... for example, using a different learning rate scheduler might lead to different conclusions". This is a fair concern, and we have run a set of additional comparisons, confirming that the proposed trend holds under varying conditions.
>
> First, we ablate our findings using two additional learning rate schedules. We show cosine decay (used in the paper), and also show results using linear decay, as well as a constant learning rate without decay.
>
> | Optimizer | Val Loss (Cosine) | Val Loss (Linear) | Val Loss (Constant) |
> |-----------|-------------------|-------------------|---------------------|
> | Signum    | 3.006             | 2.995             | **3.146**           |
> | Adam      | **2.982**         | **2.985**             | 3.158               |
> | SPlus-100 | 2.962             | 2.961             | 3.129               |
> | SOAP-100  | **2.946**         | **2.942**         | **3.089**           |
> | Muon      | 2.964             | 2.961             | 3.130               |
> | Adamuon   | **2.950**         | **2.950**         | **3.100**           |
>
> Next, we also aim to show that our findings generalize across objectives. For this purpose, we consider a **diffusion modelling** benchmark, specifically, we train a latent diffusion model on Imagenet, using a patch size of 2, following the settings in [1].
>
> | Optimizer | Val Loss (Language Modelling) | Val Loss (Diffusion) |
> |-----------|----------|----------|
> | Signum    | 3.006             | 0.7902   |
> | Adam      | **2.982**         | **0.7890**   |
> | SPlus-100 | 2.962             | 0.7839   |
> | SOAP-100  | **2.946**         | **0.7816**   |
> | Muon      | 2.964             | 0.7813   |
> | Adamuon   | **2.950**         | **0.7800**   |
>
> **In both new settings (different learning rate schedules, different objectives)**, we confirm that our argued trends hold. Spectral-normalized methods (SOAP/Muon families) outperform elementwise methods (Adam), and variance-adaptive variants (SOAP vs SPlus), (Adamuon vs Muon) are superior.
>
> **Sweeping Adam's epsilon**.
>
> To address this concern, we have performed an additional sweep of the Adam epsilon parameter:
>
> | Optimizer                  | Valid Loss |
> |-------------------------|-------|
> | SOAP                    | 2.946 |
> | Adam Eps=1e-5           | 2.995 |
> | Adam Eps=1e-6           | 2.995 |
> | Adam Eps=1e-7           | 2.983 |
> | Adam Eps=1e-8 (Default) | 2.984 |
> | Adam Eps=1e-9           | 2.984 |
> | Signum                  | 3.006 |
>
> We find that the Adam epsilon is not particular sensitive -- adjusting this term generally influences final validation loss by 0.005, which is within the standard error for our conclusions.
>
> **Clarifying the "singular value" plot**.
>
> Thank you for the feedback on this plot (Fig 3). In this plot, we examine a single dense layer, viewing its update in matrix form (e.g. a matrix of size (d_in, d_out). We take the singular value decomposition of this matrix, and examine the spread of its singular values. An accurate spectral normalizer should result in singular values that are $\pm 1$. We show that Muon achieves this accurately, whereas SOAP results in a wider spread (the ratio of maximum/average singular value is around 4). We admit the current figure may be confusing, and will clarify this in a revision version.
>
> **On minor questions**.
> - Nonstandard Adam optimizer parameters are chosen via a hyperparameter sweep using Adam as the main optimizer.
> - We will update the revision to include a summary of each method; a minimal implementation of each is also included in the project repo linked in the appendix (https://anonymous.4open.science/r/matrix-whitening-submit-5DF2/)
> - We limit our study to matrix-whitening optimizers which share a similar underlying structure (sqrt-inverse of E[gg^T]). In contrast, natural gradient-style gradients use the *fulll inverse* of E[gg^T]. These methods are less used in the current day due to their higher instability, although this is a nice direction for future work.
> - We are using weight decay performed *independently* of the optimizer, as is standard practice.
> - Validation loss is the standard to report in prior work. At sufficient scale, training loss and validation loss tend to follow closely -- in our experiments, the relative train/validation losses between optimizers is consistent.
>
> **We believe we have answered the three key concerns you have mentioned in your review**. The main concern on the generalizability of our findings has been shown via replications with a different learning-rate schedules and a different objective entirely (diffusion modelling). If you agree, would you consider updating your score? We are happy to answer any additional concerns.

---

> > ### Comment · Reviewer_HjvJ · 2025-11-25
> >
> > Thanks for your additional experiment with a diffusion model, which slightly strengthens the generalizability of the findings. I raised my score accordingly. However, I am skeptical that you were able to perform the hyperparameter sweep as thoroughly as the initial experiments, given the limited time.

---

### Official Review · Reviewer_kdfp · 2025-11-01

**Soundness:** 3
**Presentation:** 3
**Contribution:** 2
**Rating:** 2
**Confidence:** 3

**Summary:**

This paper provides a systematic decomposition and empirical analysis of matrix whitening optimizers. The authors find that both spectral normalization and variance adaptation are indispensable components. Further ablation studies show that the lookahead approximation is less effective than the low-rank approximation.

**Strengths:**

The main strength of this paper lies in its thorough empirical analysis of recently popular matrix whitening optimizers, including Shampoo, SOAP, and Muon. Through carefully controlled experiments, the authors disentangle two key factors, spectral normalization and variance adaptation, and find that both lead to significant performance improvements compared with their element-wise and non-adaptive counterparts. The experimental results are comprehensive and well support their findings, and the presentation is generally clear.

**Weaknesses:**

Although this paper presents detailed empirical analyses, it does not appear to offer important new findings. The authors only verify the effectiveness of spectral normalization and variance adaptation under a single setting, results that are already well known. Specifically, Adam corresponds to variance adaptation, while spectral normalization is employed in methods such as Shampoo and Muon. Numerous prior works have already demonstrated that these techniques improve performance across a wide range of tasks. Moreover, combining the two is not new either; the SOAP and AdaMuon papers have extensively shown that integrating spectral normalization with variance adaptation is highly effective. Therefore, it is unclear what new insights the authors provide. If the contribution is merely a systematic validation of previously established findings under a single setting, it would be difficult for such a study to be accepted at a venue like ICLR.

**Questions:**

The paper’s main conclusion, that both spectral normalization and variance adaptation contribute to the success of matrix-whitening optimizers, appears consistent with prior findings from works such as SOAP and AdaMuon. Could the authors clarify what new understanding or insight their analysis brings beyond these established results?

Could the authors provide more empirical evidence to explain how matrix-whitening optimizers work, rather than only reporting validation losses?

---

> ### Author Response · Authors · 2025-11-18
>
> Thank you for your detailed response. We address concerns below:
>
> **On the contributions of the paper**.
>
> We would like to emphasize that our main goal in this paper was to systematically show that spectral-normalization is *not the full picture*. For this reason, we view distinct optimizer families as sharing the same matrix-whitening objective, where it is immediately apparent (and empirically confirmed in our experiments) that Muon does not implement the variance-adaptive behavior of the whitening metric. While you mention "combining the two is not new", we believe that this is an important ingredient that the field has historically overlooked -- see for example the recent papers [1], [2], and the large-scale billion-parameter Muon-based [3], which all propose replacing Adam with Muon-style orthogonalization, but fail to re-introduce variance adaptation in any form. This paper aims to correct such overlookings.
>
> **On relation to prior work.**
>
> We emphasize that Adamuon and additionally NorMuon [4] are proposed concurrently -- both papers are also up for review in exactly this same ICLR conference. We opted to refer to our variance-adaptive Muon ablation as "Adamuon" to avoid introducing an additional name to a similar algorithm. (Note there is a slight difference in implementation between our “Adamuon” and the concurrently proposed Adamuon and NorMuon optimizers -- the concurrent Adamuon utilizes an additional “sign” operation, and NorMuon utilizes a column-average). We also note that the arxiv reports for both Adamuon and Normuon *do not perform a sweep over learning rates*, which we believe as limiting their generalizability, especially as variance-adaptation changes the relation between learning rate and the absolute scale of the update -- in contrast, we have performed an extensive validation, as well as generalized these findings to not only Muon but the entire class of matrix-whitening optimizers.
>
> The SOAP paper does not make explicit claims about variance adaptation, as its main comparison (Shampoo) also involves variance normalization. Rather, our work presents the first ablation of exactly the variance-adaptive component via a comparison to SPlus, which utilizes signed descent.
>
> We also note that to our best knowledge, this work is the first that ablates the various *memory-saving approximations* to variance-adaptation in the context of matrix-whitening optimizers. Our findings in Table 3 highlight that we can utilize factorization to dramatically reduce memory requirements, which is directly applicable to algorithms such as SOAP.
>
> **Experimental breadth**.
> It is a fair criticism that our experiments only considered a language modelling setting, although we note that this is standard practice in optimizer comparisons. To further strengthen our study, we have run an additional set of comparisons on **diffusion modelling**, specifically, latent diffusion modelling on the Imagenet dataset. We generally follow the setup in [5] -- we use the stable diffusion VAE, train with a patch size of 2, and utilize a flow-matching loss. We confirm that the argued trends continue to hold:
>
> | Optimizer | Val Loss (Language Modelling) | Val Loss (Diffusion Modelling) |
> |-----------|----------|----------|
> | Signum    | 3.006             | 0.7902   |
> | Adam      | **2.982**         | **0.7890**   |
> | SPlus-100 | 2.962             | 0.7839   |
> | SOAP-100  | **2.946**         | **0.7816**   |
> | Muon      | 2.964             | 0.7813   |
> | Adamuon   | **2.950**         | **0.7800**   |
>
> We hope these points clear up the contributions of this paper in relation to the prior work. Please let us know if there are additional clarifications or confusions.
>
> [1] Kwangjun Ahn, Byron Xu, Natalie Abreu, and John Langford. Dion: Distributed orthonormalized updates. arXiv preprint arXiv:2504.05295, 2025.
>
> [2] Tim Tsz-Kit Lau, Qi Long, and Weijie Su. Polargrad: A class of matrix-gradient optimizers from a unifying preconditioning perspective. arXiv preprint arXiv:2505.21799, 2025.
>
> [3] Liu, Jingyuan, Jianlin Su, Xingcheng Yao, Zhejun Jiang, Guokun Lai, Yulun Du, Yidao Qin et al. "Muon is scalable for LLM training." arXiv preprint arXiv:2502.16982 (2025).
>
> [4] Li, Zichong, Liming Liu, Chen Liang, Weizhu Chen, and Tuo Zhao. "NorMuon: Making Muon more efficient and scalable." arXiv preprint arXiv:2510.05491 (2025).
>
> [5] Frans, Kevin, Sergey Levine, and Pieter Abbeel. "A Stable Whitening Optimizer for Efficient Neural Network Training." arXiv preprint arXiv:2506.07254 (2025).

---

> > ### Comment · Reviewer_kdfp · 2025-11-25
> >
> > Thank you for the authors’ response. I am still unsure about the scope of the main contribution of this work. Are the authors claiming that adamuon (Adam + muon) as proposed in this paper is an independent line of work from the other adamuon paper? If so, I have no objection to the experiments being limited to language modeling tasks, but using only GPT-2 Base (162M) is too small in scale. I notice that the other adamuon paper conducts experiments on multiple models and datasets, up to the billion-parameter level. Otherwise, if the authors’ main contribution is the more fine-grained ablation study, I do not believe this is sufficient for acceptance at a top-tier conference.

---

> > > ### Author Response · Authors · 2025-12-02
> > >
> > > Thank you for your response. This paper attempts to study individual optimizer *components*, rather than distinct specific algorithm. Whenever a combination of components resembles a past optimizer, we refer to it with that past name (see Table 3). For this reason, we refer to a proposed variance-adapted version of Muon as "Adamuon", in reference to an arxiv preprint of the above submission. However, we note that our considered "Adamuon" is slightly different than the concurrent Adamuon (which uses a "sign" transform) and also different than the concurrent Normuon (which uses a column-averaging)."
> > >
> > > Additionally, our intent with this paper is *not* to introduce a new method, but a unification of past optimizers along with a *specific scientific finding* -- that variance-adaptation is an important component, which is often overlooked in past works.
> > >
> > > We acknowledge the limitation in the original report on a single objective, and we have shown above that these findings generalize to the widely-used Imagenet diffusion modelling benchmark as well.

---

### Official Review · Reviewer_JNF4 · 2025-11-01

**Soundness:** 2
**Presentation:** 2
**Contribution:** 1
**Rating:** 2
**Confidence:** 5

**Summary:**

This paper presents a systematic deconstruction of modern "matrix-whitening" optimizers. The authors' central thesis is that the success of these methods relies on two key, decoupled components: (1) spectral normalization and (2) variance adaptation.

Through a series of controlled experiments on a 162M parameter GPT-2 model, the paper argues that while spectral normalization (the "whitening" part) is beneficial, variance adaptation (the "Adam-like" part) is an "overlooked" and "crucial ingredient" that is "roughly as important as the spectral-normalizing aspect". The paper's primary conclusion is that variance-adapted optimizers (like Adam, SOAP, and AdaMuon) consistently and significantly outperform their sign-descent counterparts (like Signum, SPlus, and Muon).

**Strengths:**

The paper's primary strength lies in its attempt to create a controlled and rigorous experimental setup. The methodology isolates the optimizers to only the dense parameters of the Transformer and relies on a thorough, independent hyperparameter sweep for each method, which the authors use to argue against improper tuning as a confounding factor.

**Weaknesses:**

1. A Questionable Premise: The paper groups historically distinct optimization strategies—namely adaptive regularization methods (Adam, Shampoo) and spectral descent methods (Muon)—into the same "matrix-whitening" bucket. This re-interpretation, which frames all methods as approximations of a single "whitening metric" defined in Equation (2), is a non-standard premise that is not sufficiently justified or defended. Equation (2) computes a value different from adapative regularization methods which use gradient accumulation over optimization steps.

2. Unconvincing Baseline Performance (Shampoo): The paper's central argument for the importance of variance adaptation is severely weakened by its own experimental results for Shampoo. If variance adaptation is a "crucial ingredient," one would expect Shampoo (a variance-adapted method) to significantly outperform Muon (which the paper frames as a sign-descent method). However, the paper's own results show Shampoo-10 (Val Loss 2.963) is only negligibly better than Muon (Val Loss 2.964), and Shampoo-100 "fails to converge".
This strongly suggests the Shampoo baseline was not properly tuned. The authors' admission that they "disregard auxiliary design choices in each algorithm (e.g. learning rate grafting...)" all but confirms this. While this decision was made to isolate "core" behavior, it appears to have crippled a key baseline.

3. Confusing Experimental Comparisons: The analysis in Section 5, which compares SOAP to Muon, is confusing. A more natural and direct comparison to isolate the effect of variance adaptation would have been between SOAP and SPlus, as both operate on the same rotated eigenbasis. The paper's choice to instead compare SOAP (explicit eigendecomposition) with Muon (implicit Newton-Schulz) to critique the accuracy of spectral normalization obscures the main argument.


4. Insufficient Explanation for the Central Claim: The paper's primary takeaway—that variance-adapted optimizers (Adam, SOAP) perform better than their sign-descent counterparts (Signum, SPlus)—is presented as a major finding. However, this is a not so surprising observation. The major concern is that while the paper shows variance adaptation is important, it fails to provide a deep, novel explanation for why.

5. Minor Errors: The abstract states that matrix-whitening serves "two purposes" but does not clearly enumerate them in the following text, Validation loss 0.4 --> 0.04 in discussion section.

**Questions:**

Did you try grafting the Frobenius norm of Muon to Shampoo?

---

> ### Author Response · Authors · 2025-11-18
>
> Thank you for your detailed comments. Please see our response to your concerns below:
>
> **On the premise of the paper**.
>
> A main goal in this work is to view various optimizer families, which have been historically seen as "historically distinct optimization strategies" (as stated in your comments) in a unified light to *isolate precisely what influences performance*. For this reason, we intentionally adopt the view of the whitening metric as a shared goal for these optimizers. While not the default view, this perspective has been taken in prior works [1] [2] [3]. (This is also why we do not inherent techniques such as grafting, which obscure the relation between optimizers).
>
> In fact, we believe that our experiments are what justify taking this viewpoint, and is a main contribution of the paper. In the unified whitening-metric perspective, it is immediately apparent (and empirically confirmed in our experiments) that Muon does not implement the variance-adaptive behavior of the whitening metric. For this reason, we argue that "optimizers that focus solely on orthogonalizing updates will gain from re-implementing variance-adaptation in some form" (Section 6). While you mention "this is not a so surprising observation", we believe that this is an important ingredient that the field has historically overlooked -- see for example the recent papers [4], [5], and the large-scale billion-parameter Muon-based [6], which all propose replacing Adam with Muon-style orthogonalization, but *fail to re-introduce variance adaptation in any form*. This paper aims to correct such overlookings.
>
> To further support our claim on the importance of variance-adaptation, we have additionally run further comparisons on settings that adjust the learning rate schedule, as well as showing performance on diffusion modelling. (See response to reviewer veBi)
>
> **Confusing Shampoo Baseline**.
>
> It is a fair point, and it's true that the Shampoo comparison is confusing. It has been shown in [1] that Shampoo without learning-rate grafting is unstable when cached for 100 iterations, even when extensively tuned -- this is a byproduct of the stale preconditioner. **Shampoo is not a main comparison, it is SOAP-100/SPlus-100 that are the relevant comparisons**. We show Shampoo (as well as SOAP-10/SPlus-10) as a point of reference -- but we admit this is confusing for readers, and we will move these studies to the Appendix in a revised version. (Note prior works such as [2], [7], which do not even attempt a Shampoo comparison, and only use SOAP which is known to be more performant and stable).
>
> **On comparing SOAP vs Muon**.
>
> We compare SOAP to Muon as a motivating example of the contradiction when adopting spectral normalization as the goal of optimizers -- SOAP achieves the highest performance, but Muon achieves the most accurate spectral normalization. That said, SOAP vs. SPlus is indeed the most direct ablation, and we do include this comparison in Figure 3 (left) and explore the ablations carefully in Table 3.
>
> **Minor errors**.
> Thank you for pointing these typos out. We will fix these issues in the next revision.
>
> We hope these points clear up the purpose of this paper in relation to the field, and clears up the miscommunication of the Shampoo baseline. Thank you for your detailed feedback -- if there are still doubts, please let us know.
>
> [1] Frans, Kevin, Sergey Levine, and Pieter Abbeel. "A Stable Whitening Optimizer for Efficient Neural Network Training." arXiv preprint arXiv:2506.07254 (2025).
>
> [2] Vyas, Nikhil, Rosie Zhao, Depen Morwani, Mujin Kwun, and Sham Kakade. Improving SOAP using iterative whitening and Muon. 2025.
>
> [3] Bernstein, Jeremy, and Laker Newhouse. "Old optimizer, new norm: An anthology." arXiv preprint arXiv:2409.20325 (2024).
>
> [4] Kwangjun Ahn, Byron Xu, Natalie Abreu, and John Langford. Dion: Distributed orthonormalized
> updates. arXiv preprint arXiv:2504.05295, 2025.
>
> [5] Tim Tsz-Kit Lau, Qi Long, and Weijie Su. Polargrad: A class of matrix-gradient optimizers from a
> unifying preconditioning perspective. arXiv preprint arXiv:2505.21799, 2025.
>
> [6] Liu, Jingyuan, Jianlin Su, Xingcheng Yao, Zhejun Jiang, Guokun Lai, Yulun Du, Yidao Qin et al. "Muon is scalable for LLM training." arXiv preprint arXiv:2502.16982 (2025).
>
> [7] Wen, Kaiyue, David Hall, Tengyu Ma, and Percy Liang. "Fantastic pretraining optimizers and where to find them." arXiv preprint arXiv:2509.02046 (2025).

---

### Meta-Review · Area_Chair_cFJe · 2026-01-04

**Summary:**

This paper considers a systematic deconstruction of modern "matrix-whitening" optimizers. The central conclusion is that the success of these methods relies on two independent components: spectral normalization and variance adaptation. Although this paper presents detailed empirical analyses, it does not appear to offer important new findings. Therefore, it is unclear what new insights the authors provide. If the contribution is merely a systematic validation of previously established findings under a single setting, it would be difficult for such a study to be accepted at a venue like ICLR.

**Reviewer Concerns:**

Review kdfp concerns the importance of the submitted paper, and I agree with that "although this paper presents detailed empirical analyses, it does not appear to offer important new findings." I find the rebuttals cannot well address this concern. Therefore, it would be difficult for such a study to be accepted at a venue like ICLR.

**Reviewer Scores:**

I appreciate the authors' efforts to thoroughly address the rebuttal concerns, and I think some reviewers would increase their scores. Nevertheless, the expected scores still do not meet the ICLR acceptance threshold.

---

### Decision · Program_Chairs · 2026-01-26

Reject